# Denitrifying bacteria respond to and shape microscale gradients within particulate matrices

Steven Smriga[1,2], Davide Ciccarese[1,2] & Andrew R. Babbin [1✉]

Heterotrophic denitrification enables facultative anaerobes to continue growing even when limited by oxygen ($O_2$) availability. Particles in particular provide physical matrices characterized by reduced $O_2$ permeability even in well-oxygenated bulk conditions, creating microenvironments where microbial denitrifiers may proliferate. Whereas numerical particle models generally describe denitrification as a function of radius, here we provide evidence for heterogeneity of intraparticle denitrification activity due to local interactions within and among microcolonies. *Pseudomonas aeruginosa* cells and microcolonies act to metabolically shade each other, fostering anaerobic processes just microns from $O_2$-saturated bulk water. Even within well-oxygenated fluid, suboxia and denitrification reproducibly developed and migrated along sharp 10 to 100 μm gradients, driven by the balance of oxidant diffusion and local respiration. Moreover, metabolic differentiation among densely packed cells is dictated by the diffusional supply of $O_2$, leading to distinct bimodality in the distribution of nitrate and nitrite reductase expression. The initial seeding density controls the speed at which anoxia develops, and even particles seeded with few bacteria remain capable of becoming anoxic. Our empirical results capture the dynamics of denitrifier gene expression in direct association with $O_2$ concentrations over microscale physical matrices, providing observations of the co-occurrence and spatial arrangement of aerobic and anaerobic processes.

[1] Department of Earth, Atmospheric & Planetary Sciences, Massachusetts Institute of Technology, Cambridge, MA, USA. [2] These authors contributed equally: Steven Smriga, Davide Ciccarese. ✉email: babbin@mit.edu

Nitrogen (N) is a key element to organismal growth, and its availability limits biological production in many natural and engineered ecosystems[1–3]. The nitrogen cycle comprises several dissimilatory transformation processes, and among them is denitrification, the enzyme-driven stepwise reductions of nitrate ($NO_3^-$) → nitrite ($NO_2^-$) → nitric oxide (NO) → nitrous oxide ($N_2O$) → dinitrogen ($N_2$). Several key biological, biochemical, and physical constraints govern the denitrification process[4]. Among those, in the absence of molecular oxygen ($O_2$), denitrifying heterotrophs can facultatively switch their metabolisms to oxidize organic carbon with nitrogen-oxide compounds as alternate electron acceptors to $O_2$. While denitrification is routinely considered an important metabolism in fully anaerobic systems like sediment subsurfaces[5] and ocean oxygen minimum zones[6], organic particles may permit denitrification to occur within aerobic systems, expanding the niche beyond these confined environments[7].

Many bacteria proliferate in surface attached microbial communities embedded in matrices of extracellular polymeric substances[8]. Moreover, soft aggregates in hydrated environments provide resources and microscale niches to sustain microbial communities and thus play a fundamental role to maintain key ecological processes. For instance, soils[9], wastewater systems[10], and marine ecosystems[7,11] all contain ubiquitous surface attached microbial communities providing spatial niches that harbor denitrification in hydrated matrices[12]. Unlike a planktonic free-living lifestyle in which nutrient resources can be readily resupplied, microbial cells within a physical matrix are fixed in place[13], and resource availability may be diffusion-limited and sensitive to consumption[7,14]. Similarly, biofilms in the environment and in association with animals, e.g., mucoidal layers adhered to the human respiratory tract[15], can enable prolific denitrification in otherwise well-oxygenated environments. In a marine context, recent evidence has shown that coral-generated mucus harbors active denitrifiers even among tropical reefs a few meters below the atmospheric interface[16]. In several ecosystems, bacterial colonies grow subject to three-dimensional mechanical confinement whereby a colony's growth will modify its local physicochemical environment that may subsequently affect interactions with neighboring colonies[17].

Although the metabolic process fundamentally requires suboxia on the molecular scale[4], denitrification may occur in particles within fully oxygenated fluids[7,18], where local $O_2$ depletion and suboxic microzones develop due to high microbial aerobic respiration[19,20]. Microbial denitrifiers within suboxic microzones likely control the different enzymatic modular steps of denitrification dependent on the local conditions. For example, as $O_2$ concentrations decrease, cells may upregulate Nar, the nitrate reductase enzyme that catalyzes $NO_3^-$ reduction, and subsequently Nir, the nitrite reductase enzyme that catalyzes $NO_2^-$ reduction, as $NO_2^-$ becomes available and $O_2$ conditions approach anoxia[21]. The phenotypic balance among Nar, Nir, and the other enzymatic steps may drive the production and consumption of fixed nitrogen ($NO_x^-$) in the particle, with potential for microscale accumulation of nitrogen intermediates within the matrix[20]. Microbial modifications of the chemical landscape may create niches that benefit an individual cell while impacting neighboring cells and colonies competing for the same resources over just a few micrometers (i.e., at the scale of the cells themselves)[22,23]. The resulting heterogeneities can be exacerbated when pore networks are partially saturated with water.

There remains a limited understanding of the biotic underpinnings and abiotic conditions that dictate the spatial response and differentiation of the denitrification pathway within hydrated particle matrices. Nutrient availability in bulk fluid surrounding particles, particularly $O_2$, $NO_3^-$, and organic carbon, act as important controls on denitrification at the scale of a whole particle[7], but these stimuli may extend to the scale of individual microbes[24,25]. Spatial distribution and growth behavior of bacteria within a particle may affect the onset of anoxia and the activation of anaerobic metabolisms, including denitrification[26]. Microzones distinguished by key chemical and biological features[13,27,28] may be instrumental in controlling the release of metabolites into the bulk environment, and could strengthen models that predict the effect of particles on geochemical dynamics in ecosystems[7,14]. Additionally, whereas small particles prone to rapid equilibration with bulk surroundings typically dominate size spectra in number, large particles dominate surface area, volume, and mass[29,30], thus providing a common diffusion-limited habitat for anaerobic metabolisms. To evaluate the feasibility and dynamics of particle-associated denitrification, we devised a reductionist experimental system that utilizes a model denitrifier growing within a simple hydrated matrix.

We employ genetically engineered strains of the bacterium *Pseudomonas aeruginosa* wild-type PAO1. *P. aeruginosa* is found widely in the natural environment, across soils[31], estuarine[32], and marine systems[33]. Further, it is an ideal model organism representing a ubiquitous genus of Proteobacteria. In addition to its environmental relevance, *P. aeruginosa* is most commonly known as an opportunistic human pathogen known to thrive in thick biofilms such as the cystic fibrosis lung, where it can denitrify[34]. Similar studies[35] have used this organism to test the development of anoxia within model particles[36] and to report the biological response of nitrate and nitrite reductase expression[35]. Like similar heterotrophic facultative denitrifiers, it respires nitrogen oxyanions (e.g., $NO_3^-$ and $NO_2^-$) in lieu of $O_2$ under suboxic conditions. Our results reveal prominent gene expression spatial heterogeneity driven by microscale features and mechanisms that may be common for hydrated particle-based denitrification under varying bulk oxygen and nutrient regimes, which have cross-cutting utility for diverse systems.

## Results

Nitrate (Nar) and nitrite (Nir) reductase expression in PAO1 were quantified with fluorescently tagged promoter fusions (Supplementary Fig. 1), green for the NarK subunit (NarK-GFP) and red for the NirS subunit (NirS-dsRed)[35]. As the first two steps in the denitrification pathway, expression of these genes indicates a metabolic switch from aerobic respiration with $O_2$ to anaerobic respiration via denitrification. These PAO1 reporter strains were grown embedded within 3 mm diameter[14] agarose particle discs held within a custom-built gastight millifluidic device (Fig. 1; Supplementary Methods), permitting continual lateral nutrient supply of rich nutrient media (with $O_2$ and $NO_3^-$) from the bulk media via diffusion while maintaining a constant boundary condition at the particle periphery (Fig. 1a, b). This created an analog of a particle replete with dissolved organic nutrients whereby the agarose acted as an inert polymeric matrix rather than as a carbon source. Over ~24 h of growth in Luria-Bertani Broth (LB) media supplemented with $NO_3^-$, PAO1 formed densely packed stationary microcolonies (Supplementary Video 1), similar to its growth morphology in model alginate beads[15,36] and agar blocks[34]. Seeding bacterial cells within a 1% agarose matrix allows colony expansion due only to growth (passive movement) and prevents active aerotactic and chemotactic movement driven by flagellar motility[37]. A subset of agarose particles contained biocompatible $O_2$ nanosensors that could faithfully report $O_2$ conditions varying dynamically over the scale of minutes (Supplementary Fig. 2).

PAO1 readily established anoxia within particles maintained in air-saturated bulk fluid. To determine this, agarose particles

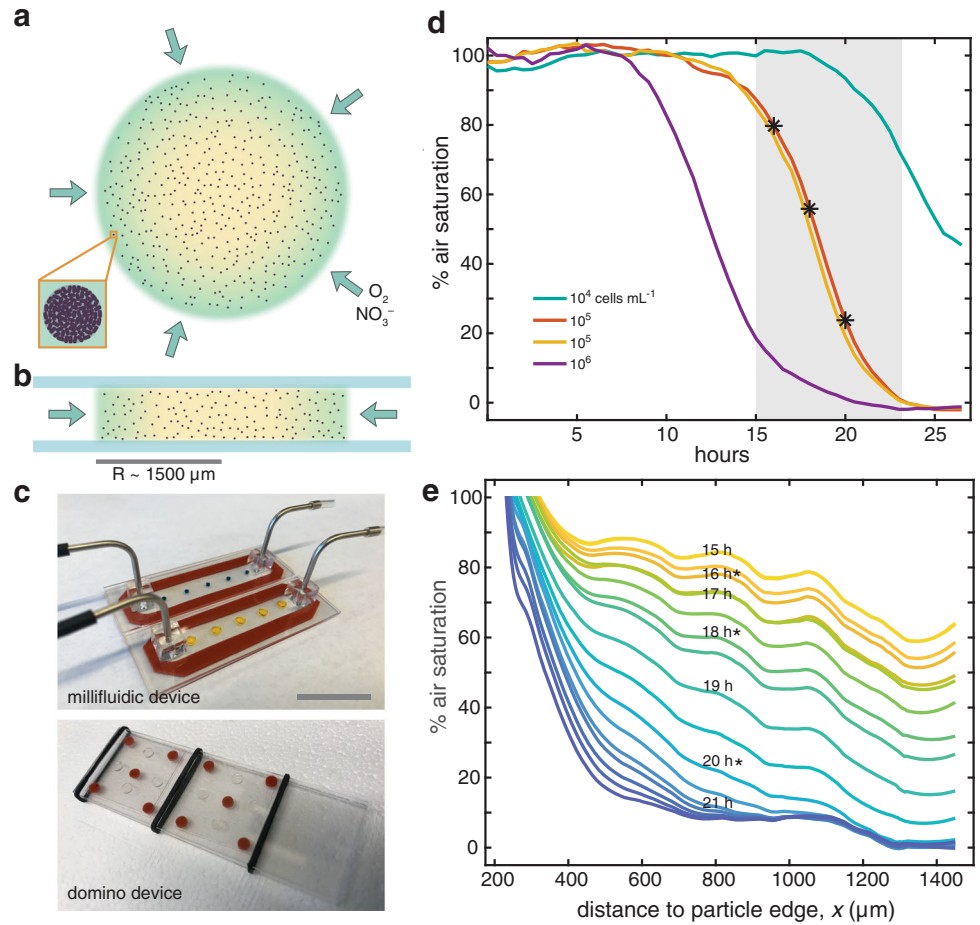

**Fig. 1 Denitrifiers create anoxia within particles in fully aerated fluid.** Agarose disc particles seeded with *P. aeruginosa* PAO1 were incubated with media in glass devices that permit only lateral diffusion into particles. **a** Topview and **b** sideview illustrating nutrient diffusion into particles as occurred in **c**, a millifluidic device with directional flow of air-saturated fluid, or a 'domino' device held within sealed glass bottles. Scale bar ~1.5 cm. **a** inset, PAO1 cells grew as dense microcolonies. **d** Air saturation across four particles as determined from microscopic signal of fluorescent oxygen nanosensors. Despite aerobic bulk fluid surrounding the particles, cell growth created anoxia and its onset depended on PAO1 seeding density, ranging $10^4$–$10^6$ cells mL$^{-1}$ (70–7000 cells particle$^{-1}$; the $10^5$ mL$^{-1}$ cell density was tested for duplicate particles). Gray shading indicates time range shown in **e** and asterisks coincide with noted timepoints in **e**. **e** The air saturation for radial profiles across a single particle ($10^5$ cells mL$^{-1}$ seeding) during the cellular respiration-driven transition from air-saturated to anoxic conditions. Suboxia first developed in the core ($x > 1000$ μm) then spread to the particle periphery ($x < 600$ μm). The 200 μm region closest to the particle edge was not resolved in this experiment.

containing $O_2$ nanosensors were co-seeded with cells at 3 different densities (~70–7000 cells particle$^{-1}$, or $10^4$–$10^6$ cells mL$^{-1}$) and allowed to grow over 24 h. Coinciding with bacterial growth, suboxia developed within particles, and the timing of onset depended on bacterial density (Fig. 1d). Specifically, suboxia manifested first in the highest initial cell density particle ($10^6$ cells mL$^{-1}$), followed by the intermediate ($10^5$ cells mL$^{-1}$) then the lowest cell density ($10^4$ cells mL$^{-1}$), respectively. In each case, the transition from the onset of suboxia to maximal anoxia occurred over ~12.5 h. Within a particle, suboxia rose first in the core then subsequently expanded outward to the particle periphery, such that it lagged behind the core, yet also itself eventually became anoxic (Fig. 1e). Meanwhile, microcolony sizes were inversely related to the seeding density, i.e., anoxia across the particle was equally able to manifest via both fewer larger microcolonies or more numerous smaller microcolonies.

Reduced $O_2$ concentrations alone are insufficient for active denitrification, however; $NO_3^-$ availability is essential. In fully anoxic bulk conditions conducive for denitrification, microcolony growth and expression of denitrification genes by PAO1 reflected $NO_3^-$ availability in the bulk and depended on radial distance within the particle. Particles embedded with PAO1 denitrification reporter strains ($10^6$ cells mL$^{-1}$) were incubated in anoxic LB media supplemented with $NO_3^-$ across 3 orders of magnitude (40, 400, or 4000 μM). Here too, only lateral diffusion of nutrients into the particles occurred (Fig. 1a; Supplementary Fig. 3). After 40 h of growth, denitrification gene expression and size were quantified for >$10^3$ microcolonies within each particle (e.g., Fig. 2). Both Nar and Nir expression were skewed toward the particle edge for lower bulk $NO_3^-$ concentrations (Fig. 2b, c, Supplementary Fig. 4), indicative of a low $NO_3^-$ flux reaching microcolonies in the particle core. In contrast, homogenous Nar expression occurred across all radial distances with 4 mM bulk $NO_3^-$ (Fig. 2b), which suggests $NO_3^-$ was nonlimiting for expression throughout the particle at such high bulk $NO_3^-$. Meanwhile, reduced Nir expression across the particle at 4 mM $NO_3^-$ compared with lower $NO_3^-$ treatments (Fig. 2c) indicates PAO1 preferentially reduced $NO_3^-$ over $NO_2^-$ throughout the particle under such $NO_3^-$-replete conditions. Bulk $NO_3^-$ also influenced microcolony size distributions (Fig. 2b, c), with larger microcolonies manifesting near the periphery closer to the $NO_3^-$ supply from the bulk fluid. Higher bulk $NO_3^-$ stimulated greater microcolony growth in the particle core, i.e., at the center, the mean microcolony radius $r$ was $6.4 \pm 0.1$ μm (sd) at 40 μM bulk

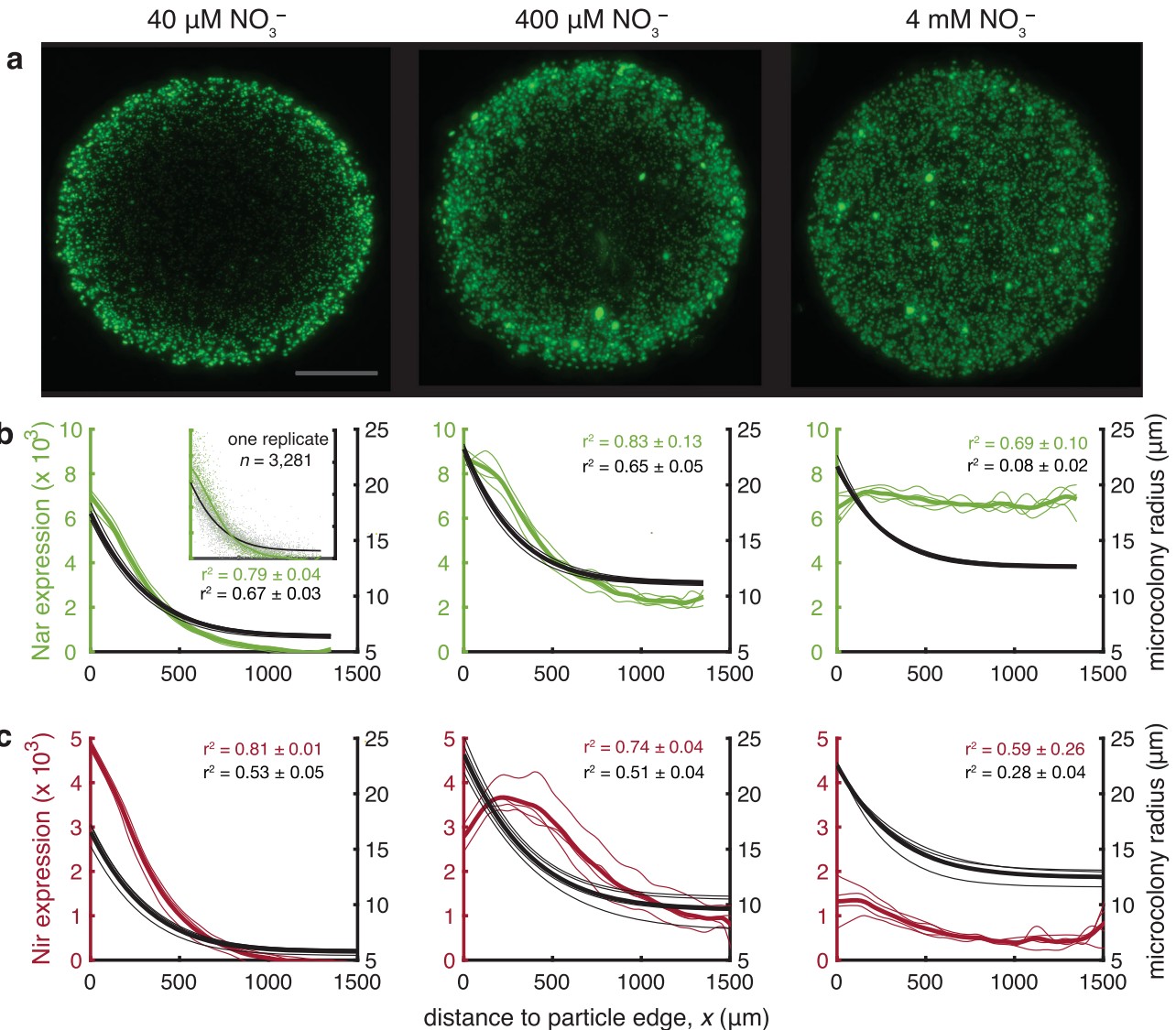

**Fig. 2 Denitrification gene expression and microcolony size within particles in fully anoxic fluid. a** Example images showing PAO1 microcolonies expressing NarK-GFP (nitrate reductase) after 40 h of growth in bulk anoxic media amended with 3 nitrate concentrations. Scale bar = 700 μm. **b** Mean relative expression of NarK-GFP for microcolonies, and mean radii of those microcolonies, shown in relationship to the radial distance to the nearest particle edge. The mean NarK-GFP expression for microcolonies in each particle (3–4 replicates) is indicated by a thin green line and the mean for all particles by a thick line. Similarly, an exponential model fit of microcolony radii for each particle is indicated by a thin black line and the mean for all particles by a thick line inset, Results for one example particle in **a**, illustrating the data for $n$ microcolonies relative to the respective fits for those data. **c** Same as **b** but for reporter strain NirS-dsRed (nitrite reductase) expression in separate particles incubated in parallel.

$NO_3^-$ but 12.7 ± 0.1 μm (sd) at 4 mM. Notably, under 4 mM bulk $NO_3^-$, size was skewed toward the particle periphery even while Nar expression was not, suggesting maximum cell-specific Nar expression rates occurred across all radial locations despite biomass production in the particle core remaining $NO_3^-$ limited.

Not only did particle denitrification readily occur in anoxic bulk fluid as expected, it was also prevalent among particles in oxygenated bulk fluid. Moreover, the spatiotemporal expression of Nar and Nir coincided with the development and microscale distribution of suboxic conditions. In these experiments, agarose particles were embedded with either NarK-GFP or NirS-dsRed PAO1 ($10^6$ cells mL$^{-1}$) as previously, then incubated in partially oxygenated LB media (50% air saturation) containing $NO_3^-$ (40 μM). A subset of particles was co-embedded with oxygen nanosensors to enable imaging of the $O_2$ landscape. Minimal Nar and Nir expression were detected for the first 14 h of incubation. Strikingly, Nar and Nir activation then occurred as a wave that

traced the initiation and expansion of suboxia (Fig. 3a–c). At the earliest stage of the wave (following 16–18 h of growth), expression increased substantially in the vicinity of the particle core (Fig. 3d), crafting a transition zone of only 390 ± 50 μm (sd) width that differentiated low- from high-expression microcolonies (Fig. 3e, Supplementary Fig. 5, Supplementary Fig. 6). This upregulation of expression coincided spatially with the development of anoxia in the core (Fig. 4, Supplementary Fig. 7; $x > 1000$ μm). During the next stage of the wave ($t = 20$–22 h), expression near the core elevated further while also initiating farther afield, widening the transition zone and reflecting expansion of suboxia toward the particle periphery. As the wave progressed ($t = 24$–28 h), expression diminished near the core, and consequently, the transition zone width contracted (200 ± 15 μm (sd)) even while its outermost edge neared to within 200 μm of the particle edge, where $O_2$ conditions had become anoxic (Fig. 4). Finally, in the final stage of the wave (exemplified

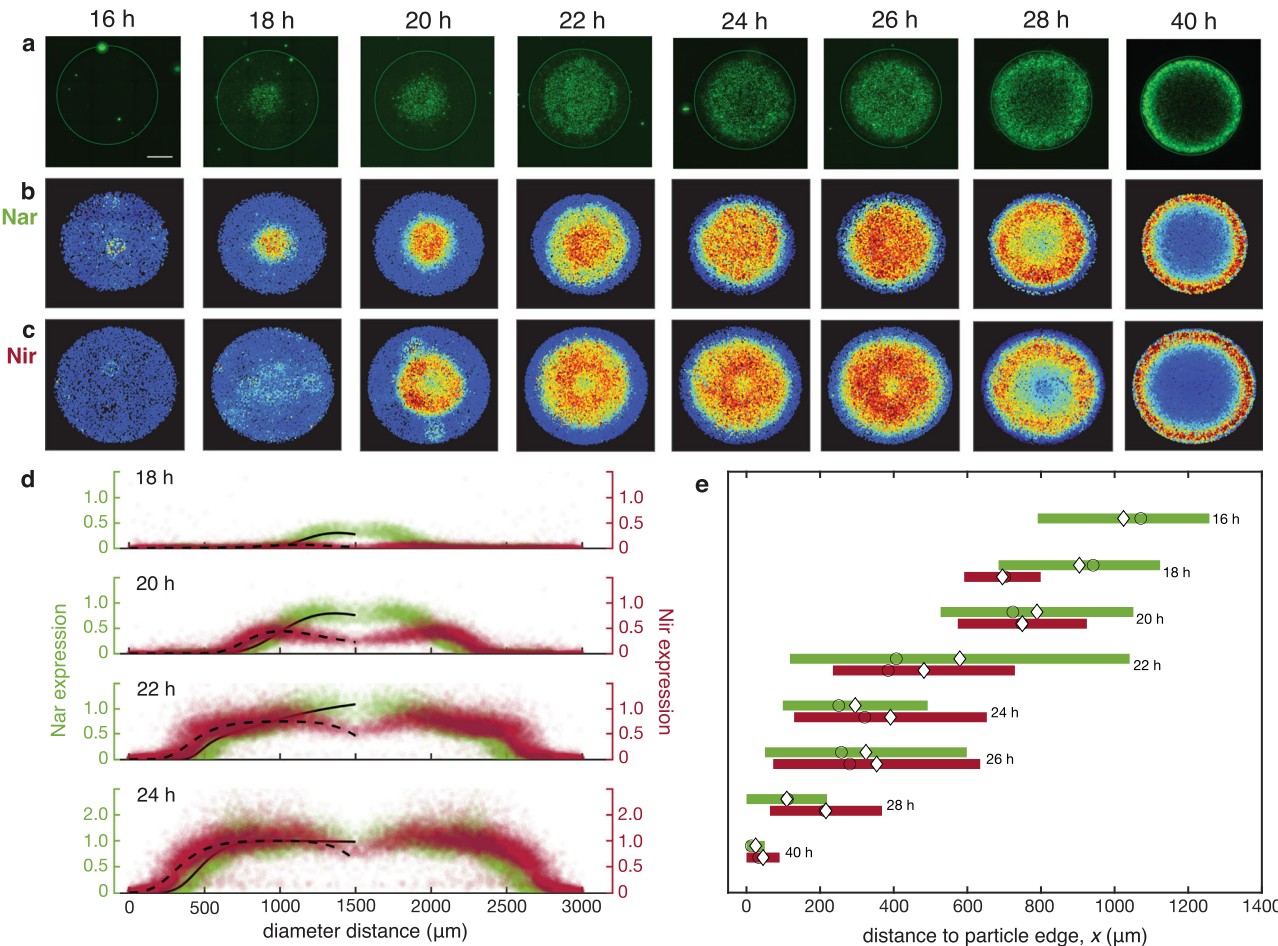

**Fig. 3 Radial migration of denitrification expression within particles in partially aerated fluid.** Particles seeded with cells were incubated in LB media saturated with 50% air and supplemented with 40 µM $NO_3^-$, then stopped at various timepoints. **a** Example images showing PAO1 microcolonies expressing NarK-GFP (nitrate reductase). Scale bar = 700 µm and applies to all images. Separate particles with the reporter strain for NirS-dsRed (nitrite reductase) were incubated in parallel. **b** Relative expression of NarK-GFP and **c** NirS-dsRed in particles. All microcolonies from 3–4 replicate particles per timepoint are represented. Nar expression initiates at the particle core while Nir expression initiates just proximal to it. For both Nar and Nir, maximal expression migrates outward creating a wave over subsequent timepoints. **d** Relative microcolony expression (as in **b** and **c**) shown as a microcolony's radial location within the particle. An expression intensity fit was calculated as mean fits of each strain at each timepoint (Supplementary Note). Here the maximum fit value for each reporter strain at $t = 24$ h was set equal to 1, and shown are the expression values for all microcolonies relative to 1. **e** The radial location and range of the transition zone for each timepoint, approximated as the sloped region closest to the particle edge in the expression profile for NarK-GFP (green) and NirS-dsRed (red) (see also Supplementary Fig. 6). The midpoint of the slope (white diamonds) and the inflection point of the slope (circles) are indicated. NarK-GFP shows significantly higher expression between timepoints for midpoints (one-way ANOVA; F 1,35 = 27.9, $p = 4.0 \times 10^{-11}$) and for inflection points (one-way ANOVA; F 1,35 = 17.2, $p = 9.5 \times 10^{-9}$). NirS-dsRed also showed significantly higher expression between timepoints for midpoints (one-way ANOVA; F 1,35 = 14.9, $p = 4.5 \times 10^{-8}$) and for inflection points (one-way ANOVA; F 1,35 = 21.0, $p = 1.0 \times 10^{-9}$).

at $t = 40$ h), high expression was confined to a fine band distantly from the core, yet not directly at the edge, as evidenced by a very narrow transition zone ($57 \pm 13$ µm (sd) width). Notably, at the last stage, $O_2$ throughout the particle was elevated above the minimal observed levels, in contrast to previous stages ($t = 24$–28 h; Fig. 4, Supplementary Fig. 7). This may reflect anoxic acclimation by PAO1 to preferentially perform denitrification over oxidative respiration thereby permitting $O_2$ to diffuse more readily through the particle matrix.

Throughout the evolution of particle suboxia, Nar and Nir activity corresponded with the estimated spatiotemporal availability of $O_2$ and $NO_3^-$ in particles. Curve fits for microcolony fluorescence signal data were generated by assuming that $O_2$ and $NO_3^-$ concentrations at each radial location were the only controlling factors on expression, wherein $O_2$ inhibits expression exponentially and $NO_3^-$ has a directly proportional relationship, i.e., $E \propto e^{-k[O_2]} \times [NO_3^-]$. Approximating the distributions of $O_2$

and $NO_3^-$ in the particle as simple logistic functions, denitrification gene expression, E, is governed by the following relationship:

$$E = \alpha \times e^{-\beta\left[2[O_2]_{\text{bulk}}\left(1-\frac{1}{1+e^{-\gamma x}}\right)\right]} \times [NO_3^-]_{\text{bulk}}\left(1 - \frac{1}{1+e^{-\delta(x-\varepsilon)}}\right)$$

whereby α, β, γ, δ, and ε are fitting parameters and $x$ is the distance to the particle edge. (Supplementary Note). The shape of this fit prediction matches the observed empirical data remarkably well (Fig. 3d, Supplementary Fig. 5), and the midpoints and inflection points from one timepoint to the next were significantly different (one-way ANOVA, F 1,8 = 27.9, $p = 4.0 \times 10^{-11}$ for Nar midpoints, F 1,8 = 14.9, $p = 4.5 \times 10^{-8}$ for Nir midpoints, F 1,8 = 17.2, $p = 9.5 \times 10^{-9}$ for Nar inflection points, and F 1,8 = 21.0, $p = 1.0 \times 10^{-9}$ for Nir inflection points). These fits illustrate how the response of PAO1 *nar* and *nir* gene expression reflects the balance between bacterial consumption and diffusion of $O_2$

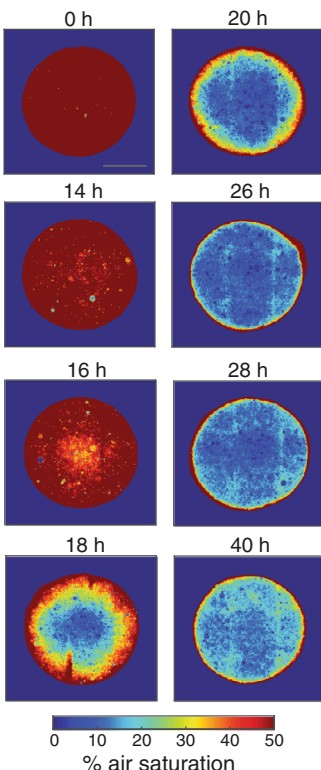

**Fig. 4 Evolution of anoxia within particles in partially aerated fluid.** Two-dimensional profiles of air saturation were generated from particles co-seeded with the NarK-GFP reporter strain (nitrate reductase) and oxygen nanosensors. Scale bar = 700 μm. Separate analogous particles with the NirS-dsRed reporter strain (nitrite reductase) were incubated in parallel (Supplementary Fig. 7). For both, suboxic conditions develop in the particle core and then migrate outward toward the particle periphery over time.

and $NO_3^-$ from bulk surrounding fluid. In this manner, the fluorescence signal diminishes after achieving its peak and microcolonies remain small behind the advancing fluorescence wave while colonies continue to expand ahead of it (Supplementary Fig. 5, Supplementary Fig. 8). Nar was downregulated in the wake of the wave, causing microcolony expansion to slow or cease in the absence of respiration. Putatively, continued $O_2$ and then $NO_3^-$ uptake by large microcolonies at the periphery created growth limitation for microcolonies in their shadows farther from the bulk fluid source. Nir expression also advanced as a wave but interestingly created an annulus of maximal expression bounded by lower expression toward both the periphery and center of the particle. This expression pattern likely reflects localized production and utilization of $NO_2^-$ within the particle interior. Exterior to the ring, ample $NO_3^-$ from the bulk favored nitrate reductase, but interior to the ring, $NO_3^-$ and $NO_2^-$ were diffusion-limited.

Importantly, the heterogeneous distribution of Nar and Nir expression across the particle also manifested among PAO1 cells at the scale of individual microcolonies. High magnification colony-scale images near the particle periphery and in the transition zone revealed a common phenotype reflecting the overall expression across the particle whereby the core of a single colony is expressive but the outer margin is not. As quantified for $>10^3$ microcolonies per particle, the subregion expressing Nar or Nir (i.e., "on") varied with distance from the particle edge (Fig. 5). A thin radial zone within the particle (distance from the particle edge, $x \sim 90–240$ μm) harbored high heterogeneity with colonies ranging from 0–100% as shown (Fig. 5b, c). This narrow transition zone aligned with that quantified at lower magnification

(Figs. 3, 4), indicative of a sharp transition from $O_2$ to nitrate- and nitrite-driven respiration. In the flanking region exterior to this zone ($x < 90$ μm), Nar and Nir in microcolonies were uniformly off. In the region interior to this zone ($x > 240$ μm), Nar was predominately on (median colony fraction expression = 0.78 ± 0.40 interquartile range; Fig. 5b) and Nir was almost completely on (median colony fraction expression = 0.95 ± 0.25 interquartile range; Fig. 5c). These expression characteristics resulted in a significantly stronger population bimodality for Nir than for Nar (Fig. 5d); Kolmogorov–Smirnov nonparametric test for probability distribution similarity, $n_1 = 1554$, $n_2 = 1354$, $p = 1.2 \times 10^{-40}$, $Dn = 0.25$. Akin to the annular feature observed at lower magnification (Fig. 3), this binary Nir expression likely reflects localized endogenous production and utilization of $NO_2^-$. Since $NO_2^-$ is not continuously supplied from bulk media via lateral external diffusion like $NO_3^-$, microcolonies in the interior use Nar to produce $NO_2^-$, which is then preferentially consumed via Nir within each microcolony as the next most available oxidant for generating energy.

Respiratory shading by exterior colonies and cells was a key emergent feature among microcolonies within the agarose particles, occurring in a fractal-like geometry. At the particle-scale ($R \sim 1500$ μm), nutrient consumption by microcolonies at the periphery prevented uptake by those at the interior; similarly, at the microcolony-scale ($r \sim 25$ μm), cells at the margin prevented uptake by those at the center, with consumption generating gradients of uptake flux at both scales. Shading did not occur for microcolonies in the particle core through the early stages of suboxia, as Nar and Nir expression were absent and microcolony size distribution was uniform over the first 14 h (Fig. 3, Supplementary Fig. 8). Since microcolonies were small over this stage, the $O_2$ flux to the center outpaced aerobic respiration, and cell biomass at the particle-scale had not yet substantially diminished diffusive $O_2$ availability. Then, owing to cell growth and the onset of shading, the $O_2$ concentrations decreased rapidly over ~2 h (Fig. 4). As such, respiratory shading should be diminished when the density of microcolonies is lower. We tested this hypothesis with particles seeded at very low density (~$10^2$ cells mL$^{-1}$) resulting in 1–6 microcolonies particle$^{-1}$. Indeed, after 40 h of growth, the resultant microcolonies were quite large ($r = 62 ± 11$ μm (sd)); Supplementary Fig. 9) regardless of radial location within the particle, reflecting low intercolony competition for oxidants that readily diffused throughout the particle. While here respiratory shading across scales occurred for a clonal population, natural multispecies communities may additionally distribute functional roles among diverse taxa, e.g., anammox aggregates spatially differentiate such that aerotolerant species encase the outer perimeter to respire $O_2$, shielding *Planctomycetes* to perform oxygen-inhibited anammox in the interior[10].

## Discussion

Our results capture to our knowledge for the first time the initiation and dynamics of denitrification gene expression, denitrifier growth, and $O_2$ distribution over the microscale space within a particle matrix. This model system can be used to investigate the spatial constraints of real-world solid systems in which denitrification is prolific as well as the microscale heterogeneity that arises as a result of biological and physical phenomena. The onset of anoxia in a particle's core coincided with the upregulated expression of nitrate and nitrite reductases (Figs. 3, 4). Moreover, the dynamic expression of Nar and Nir traced the radial expansion of anoxia to the edge of the particle (Supplementary Fig. 7). The results are broadly consistent with theoretical models that describe substrate diffusion coupled with bacterial oxic and anoxic respiration in particles[13,14,26,38,39]. A

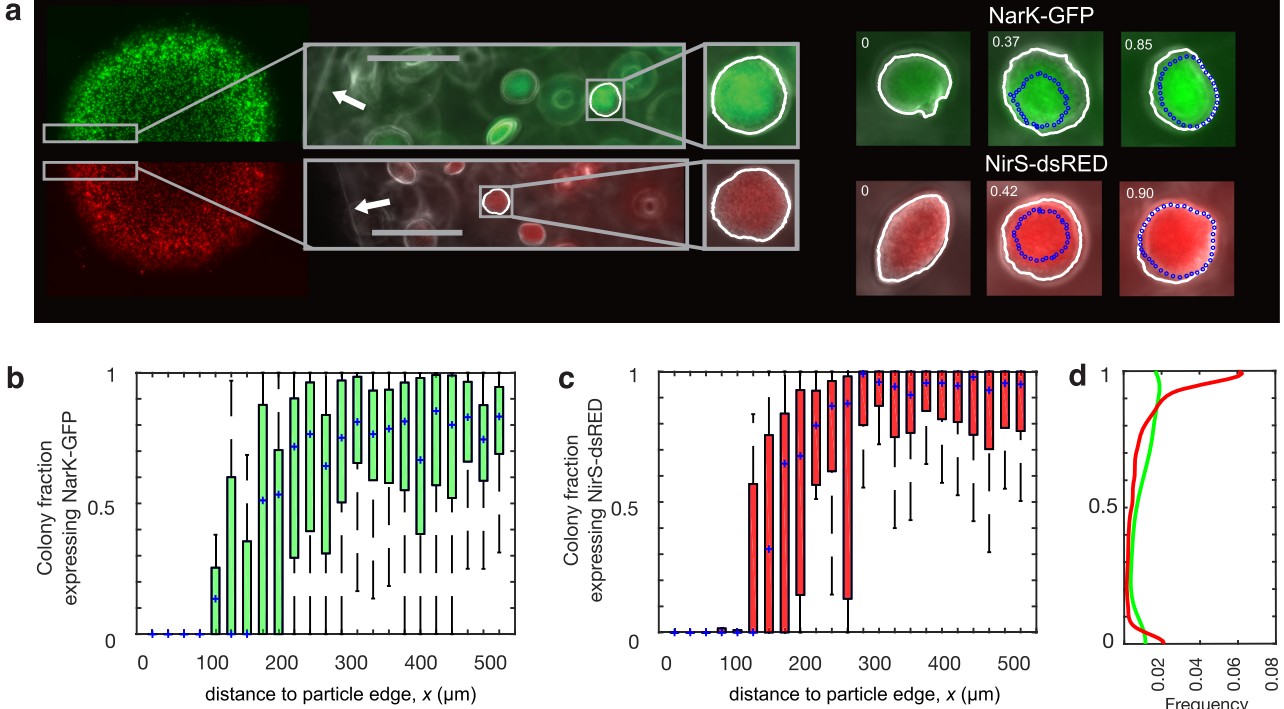

**Fig. 5 Heterogenous denitrification gene expression within individual microcolonies. a** PAO1 NarK-GFP (top) or NirS-dsRed (bottom) were grown in separate particles in LB media saturated with 50% air and supplemented with 40 µM $NO_3^-$. Arrows indicate the particle edge; scale bars = 100 µm. For hundreds of microcolonies, the fraction expressing either NarK or NirS was quantified, and example microcolonies (right) illustrate 'on' fractions ranging from 0 to 0.90, with 'on' subregions noted by blue dotted lines. **b** Microcolony fraction expression for NarK-GFP and **c** NirS-dsRed, as a function of distance to the nearest particle edge. Shown are the median (blue crosses) and quartiles binned over 25 µm of radial particle space. Colonies were primarily 'off' in the aerated zone nearest bulk fluid ($x < 100$ µm) and primarily 'on' in anoxic conditions ($x > 250$ µm). In the aerobic zone, the median expression fraction of NarK-GFP is $3.1 \times 10^{-5}$ (interquartile range $5.5 \times 10^{-5}$) whereas the NirS-dsRed expression fraction is $2.7 \times 10^{-5}$ (interquartile range $1.5 \times 10^{-5}$). Expression of both genes are not significantly different from each other (Wilcoxon rank sum; $p = 0.1$). In the anaerobic zone, the median expression fraction of NarK-GFP and NirS-dsRed are 0.70 (interquartile 0.39) and 0.94, (interquartile 0.24), respectively. These anoxic fractional expressions are significantly different from each other ($n_1 = 1160$, $n_2 = 1078$, Wilcoxon rank sum; $p = 2.2 \times 10^{-32}$, $w = 2724$). The occurrence of heterogenous partially-on microcolonies reflects a sharp transition zone between presumptive aerobic and anoxic conditions ($x \sim 100$–250 µm). **d** Probability density functions for each reporter strain indicate stronger bimodality and higher binary expression for NirS-dsRed than for NarK-GFP. The distribution of NarK v. NirS expression are significantly different from each other (two sample Kolmogorov–Smirnov test; $n_1 = 1554$, $n_2 = 1354$ $p = 1.2 \times 10^{-40}$, $Dn = 0.25$) with significantly different medians ($n_1 = 1554$, $n_2 = 1354$, Wilcoxon rank sum; $p = 1.9 \times 10^{-33}$, $w = 1.9 \times 10^6$).

frequent simplification among models is the treatment of microbes at steady-state with no net accumulation of biomass. Here we have demonstrated through direct observation that microbial growth will drive the dynamic evolution of particle anoxia given resupply of nutrients available in bulk surrounding fluid. Sharp lateral transitions characterized the spatiotemporal dynamics of denitrification (Figs. 3c, 4). When microcolonies were small, cells at maximal expression were separated from cells devoid of expression by ~400 µm, a distance that diminished to <10 µm when microcolonies were large, as evident for cells within single microcolonies (Fig. 5). Denitrification and the balance between cellular Nar and Nir expression will acclimate to the lateral $O_2$ gradient as it widens or constricts, as occurs when the particle core remains pinned to anoxia while $O_2$ concentrations vary in the bulk environment. Although sharp microscale chemical features have been quantified in particles[20,28,29,39], our approach using oxygen nanosensors provides a much finer spatial resolution and our results overlay the cellular growth and transcriptional response of denitrifiers to these features, revealing the effects of competition for resources within them.

The three bulk conditions tested here (air-saturated, partially oxygenated, and anoxic) expand our understanding of the effects of bulk resources on microscale particle denitrification. Whereas bulk $NO_3^-$ availability affects the denitrification potential at the

scale of a whole particle (Figs. 2, 3), the resultant denitrifier activity controls the release of N intermediates ($NO_2^-$, NO, and $N_2O$) from particles into the local ecosystem. Through complete denitrification to $N_2$, N intermediates should be entirely consumed locally within the particle, particularly if $NO_3^-$ supply to a particle is limited and it harbors ample labile organic carbon to support growth of dense bacterial biomass. Under particle conditions with replete organic carbon, we observed high expression of Nir within a narrow ring across three distinct experiments (Figs. 2c, 3c, 5c), putatively indicative of a phenotypic preference for complete denitrification. Conversely, if incomplete denitrification occurs, potentially due to an abundance of $NO_3^-$, N intermediates may be released into the bulk. Indeed, in experiments with replete 4 mM $NO_3^-$, high expression of Nar coincident with low expression of Nir suggests PAO1 preferentially reduced $NO_3^-$ (Fig. 2b) and released $NO_2^-$ into bulk effluent. The organisms likely opt to maximize the turnover of $NO_3^-$ to $NO_2^-$ rather than efficiently reducing $NO_3^-$ completely to $N_2$.

The results broaden our mechanistic knowledge of the physiology and physical ecology of the pathogen *Pseudomonas aeruginosa*, and by employing it as a model, we have unveiled novel aspects of bacterial denitrification as it occurs in a generalized hydrated matrix structure akin to a biofilm. For example, in a thin spatial band of the matrix, represented here by an annulus of

maximal Nar and Nir expression (Figs. 3, 5), $NO_2^-$ production and consumption are tightly coupled within an $O_2$ gradient when sufficient $NO_3^-$ is available. As *P. aeruginosa* microcolonies within matrices can develop tolerance to antibiotics[15], our results suggest that this onset of resistance may coincide with a switch to denitrification driven by microscale $O_2$ gradients when $NO_3^-$ is available. Additionally, across $O_2$ gradients, the genes for the first two denitrification steps are not necessarily described by the same distributions; Nar appears to exhibit a more uniform expression distribution whereas the duality of Nir as either fully off or fully on is pronounced (Fig. 5b–d). Moreover, the characteristics and dynamics observed here likely govern the biochemical structure that organisms experience in complex systems like marine particles, soils and sediments, wastewater granules, and microbial mats. Under persistent $NO_3^-$ limitation and anoxia in a particle, sulfate reduction or methanogenesis will be the next likeliest heterotrophic metabolisms. As such, experimental particles co-seeded with a sulfate reducer or methanogen along with PAO1 should reveal downregulated denitrification coincident with upregulated sulfate reduction or methane production in the particle core[7] along with the possible release of sulfide or methane[40]. Regardless of the specific anaerobic pathway, the net release of reduced compounds from a suboxic particle is underpinned by heterogeneous microzones, with net production of reduced compounds in some microzones and net consumption in others[27].

Here, the agarose discs permitted lateral diffusion from the bulk and smaller-scale three-dimensional diffusion across a colony, as may be common for ventilation in natural particle systems, such as biofilms, microbial mats, sediments, and marine aggregates[13]. Further, by maintaining a steady boundary condition around the perimeter of the discs, the system well-approximates a planar geometry folded upon itself. Since denitrification can occur so close to the oxygenated water interface in localized microbial pockets in this simplified empirical system (e.g., peak Nir expression at $t = 40$ h occurred 90 μm from the particle edge), naturally occurring organic-rich sediments and biofilms bounded by aerated fluid likely promote substantial denitrification within the uppermost layers, even when $O_2$ is available within the surrounding matrix at a specific depth layer[41]. Because of high cell density within a microcolony, denitrification occurs at the core even when the periphery aerobically respires. The multiple scales shown here, from individual microbe (Supplementary Fig. 1) to microcolony (Fig. 5) to particle (Figs. 2, 3), are each essential for accurately representing the metabolic diversity within a solid matrix, and models should incorporate emergent bio-physico-chemical properties from each in order to resolve the net effects for a bulk system, regardless of the specific environment. Whereas aerotaxis and other chemotactic responses were not permitted due to the tightness of the agarose matrix, future studies could investigate how motile bacteria may disrupt the emergent gradients observed here. The microscale study of particles and the stochastic heterogeneities they contain can thereby improve the resolution of large-scale features and effects of a bulk system, regardless of the specific environment.

**Reporting summary**. Further information on research design is available in the Nature Research Reporting Summary linked to this article.

## Data availability
Primary data are available as Supplementary Data Files S1–3, and upon request to the authors.

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

## Acknowledgements

We thank Masanori Toyofuku and Nobuhiko Nomura for providing the fluorescent PAO1 strains. We gratefully acknowledge Daniele Bianchi and Rémi Buisson for helpful discussions on the experiments and Soeren Ahmerkamp and Lars Behrendt for helpful discussions on oxygen nanosensors. We appreciate the comments of Benedict Borer on the manuscript. This work was supported by Simons Foundation award #622065, the MIT Ferry Fund, an MIT Environmental Solutions Initiative seed grant and startup funds to A.R.B. We additionally recognize the generous financial contributions of Dr. Bruce L. Heflinger in making this work possible.

## Author contributions

A.R.B. conceived of and supervised the study. S.S. and A.R.B. designed the experiments that S.S. conducted. S.S., D.C., and A.R.B. analyzed the data. All authors contributed to writing the manuscript.

## Competing interests

The authors declare no competing interests.
