## [Peer Review File · Communications Biology]

Reviewers' comments:

Reviewer #1 (Remarks to the Author):

The authors investigate the expression of Nir and Nar genes of *ps. aeruginosa* cells embedded in agar disks, immersed in microfluidics chambers. Fluoresce markers were inserted in the promotor region of genes, so that the gene expression could be visualized. To monitor oxygen the authors applied oxygen sensitive nanoparticles to their agar disks. With this setup, the authors performed time series experiments and could demonstrate a gradual development of expression levels of Nir and Nar inversely correlated with oxygen availability in the agar discs. The oxygen was consumed by *Ps. Aeruginosa* cells, themselves, and in this way, the cells triggered the evolution of sub-oxic sites in the discs, thereby promoting the expression of Nir and Nar genes in cells reseeded in these environment. The NAR expression preceded the Nir expression in time and in the case of high nitrate concentrations also in intensity in full agreement with results from classical culture studies. In general, I find this study very elegant designed and performed. The study shed light on the dynamics physical and biological mechanisms operating on a very small scale in particles. The work also offers an elegant technology, enabling direct in situ visualization of proxies for microbial metabolic processes. Data are convincing and the paper is well written. I recommend publication as is.

Reviewer #2 (Remarks to the Author):

Comments from the Reviewer of the Smriga et al. manuscript 'Denitrifying bacteria respond to and shape microscale gradients within particulate matrices' (Nat Comms)

Smriga et al. present a manuscript describing their investigations of the impact of oxygen levels on nitrate and nitrite reductase expression using the model pseudomonad, *P. aeruginosa* PA01 and artificial agarose particles. While the use of various polymers to capture bacteria into a 3-D structure and to test growth and phenotype response to changing conditions using microfluidics is not novel, I have found this a very interesting manuscript to read, and it adds to our greater understanding of how complex 3-D structures such as soil pore networks at the millimetre and micrometre scales effect bacterial growth and activity.

I would like three comments added to the appropriate places. First, a justification for using *P. aeruginosa* PA01 in this work needs to be added to the end of the Introduction; PA01 is not really a model for denitrifying bacteria in soils or sediments, though it is regarded as a model opportunistic human pathogen.

Second, the flagella-mediated mobility, aero and chemotaxis responses of PA01 need to be acknowledged and the impact this might have on nitrate and nitrate gene expression measurements within microcolonies considered (no independent measurements of cell densities or distributions were made, e.g. by using a third GFP-marked strain).

Third, I would like to see some reflection on how the physical structure of the agarose particles might differ from particulate matter found in soils and sediments, and from the mucosa or EPS matrix of bacterial biofilms. Agarose particles will be uniformly hydrated and will not provide varied solid surfaces for attachment as compared to soil and sediment particles which may also contain their own pore network. The agarose particles are more similar to mucosa and biofilms, but at the agarose concentrations used, are likely to limit free cell movement and compress expanding microcolonies.

I was minded that very few statistical tests results were mentioned in the text or figures to substantiate claims that there were differences in treatments. Some comments should be added to say when curves or other data were assessed qualitatively and when a statistical test was employed.

Perhaps some summary tables of mean values, infection points and final values for microcolony growth and expression levels could be provided in the Supplementary Information along with simple statistical tests.

Comments

1) Lines 18-20. It is not clear what is meant by this sentence as modelling particles (and not bacterial distributions on or in them) is conflated with the experimental observations of bacterial growth within particles (and not of particle shapes).

2) Line 26. This should be altered to 'nitrate and nitrite reductase expression' or something similar.

3) Line 27. Correct to 'develops'.

4) Introduction. When I read this, I had the overall impression that the manuscript was focussed on soils and sediments, rather than at human mucosa or biofilms, which makes *P. aeruginosa* PA10 a rather unusual choice for a test organism. Perhaps this needs to be justified towards the end of the Introduction (lines 81-83) rather than at the start of the results (lines 90-93).

5) Details. The flagella-mediated motility of PA01 cells should be noted, as should be the chemotaxis and aerotaxis responses, as these will all play some role in the distribution of cells within microcolonies as local population densities increase. Will quorum sensing also be a factor?

6) Line 98-100. Were any observations or tests made to determine whether the agarose-trapped cells were capable of moving within the agarose disks? PA01 cells are motile and can move in semi-solid agar (0.3% w/v), and cell motility will result in faster microcolony expansion; furthermore, a chemotaxis response will result in the hollowing-out of microcolonies when nutrients are reduced.

7) Line 111. Some reference to other work which defines anoxic conditions for PA01 should be provided, otherwise this statement needs to be reworded, perhaps using the expression of the nitrate and nitrite reductases as an indicator that anoxic conditions had been reached.

8) Line 118. Anoxia is not 'initiated' but rather first reached or achieved, and the anoxic region expanded, rather than anoxia itself.

9) Line 142. Avoid the use of the term 'averaged' as this lacks specificity. Change to mean if this is what is meant.

10) Line 142. What are the errors indicated here?

11) Line 142. This is the first place where means and errors are provided, but why not earlier in the descriptions of microcolony sizes and anoxic zones?

12) Lines 147-148. Please reconsider this sentence. Agarose particles were continuously washed with LB media, and cells within particles are surrounded by fluid too (but not always oxygenated) – so which is the bulk fluid?

13) Lines 152-153. Is it relevant to mention here that some of the particles were sacrificed for measurements? This should be in the Methods.

14) Line 181. I am not sure that the equation is required here and is probably more suited to the Supplementary Information.

15) Lines 185 – 187. According to the Journal Guidelines, the Degrees of Freedom should be provided

for each F statistic.

16) Lines 195-197. The lower levels of Nir expression within particles might be conflated by aero or chemotaxis of cells.

17) Line 208. Change to 'as shown (Fig. 5b-c)' or something similar.

18) Line 212. What is 'IQR'? This not explained here or in the Supplementary Information.

19) Line 215. What is the K-S test? This should be indicated in full or the acronym noted in the Supplementary Information.

20) Line 285. *P. aeruginosa* PA01 is more than an animal pathogen.

21) Lines 307-309. Surely 3-D diffusion centred on microcolonies within the disks occurred?

22) Lines 307-309. The comment regarding 1-D diffusion in biofilms is probably not defensible. While there is no agreed minimal depth (in terms of distance or the number of cell layers), even a mono-layer will establish gradients immediately above it in the laminar layer between the cells and the bulk fluid.

23) Line 310. It is unlikely that a steady boundary condition was maintained around the disks, as the downstream region would be subject to turbulence resulting from differential liquid pressure/flow.

24) Lines 316-317. Cell densities within microcolonies were not measured in this work, so care must be taken here and elsewhere to acknowledge that nitrate and nitrate reductase expression may not be directly linked to cell densities; aero and chemotaxis may have resulted in cells migrating out of the colony cores towards the edges.

25) Supplementary Methods. Where all assays of cell growth and behaviour undertaken at 37°C?

26) Lines 641-645. A comment should be added in this section to indicate that the promoter fusions had no significant effect on the down-stream expression of the nitrate and nitrite reductase genes.

27) Line 648. Regrettably both the naming and composition of LB has become a little confused, and the level of NaCl in different formulations is known to effect gene expression in *Escherichia coli* and other bacteria. I would suggest adding the formulation of LM here in parentheses or citing the supplier if pre-mixed media was used.

28) Lines 671-673. What temperature was the molten agar allowed to cool to before cells were added? Normally the molten agar would be equilibrated to 50/55°C (just above setting temperature) before use.

29) Lines 679 etc. It is annoying to have to report Imperial measurements, but here and elsewhere when it is necessary, the appropriate SI conversions should be provided.

30) No section is provided in the Supplementary Information detailing the approach taken to data analysis and comparisons. What software was used to fit and test curves? What statistical software was used for tests? Were data or residuals (e.g., from the ANOVA tests) assumed to be Normally distributed or was this tested? (Journal Guidelines require that assumptions are noted).

31) Fig. 1 Legend. The scale bar in (b) is indicates as approx. 1500 μm and not as 1.2 cm as stated.

32) Fig. 5. The graphs might be more easily interpreted if the outliers were removed. Are the outliers

incorporated into the density functions?

33) Fig. 5. The medians are very hard to see and perhaps could be indicated by thicker lines.

34) Supplementary Fig. 1 Legend. If mean signals are discussed, why a box-plots shown (normally indicating min, Q2, median, Q3 and max data)? Why are no statistical tests shown indicating whether or not treatments are significantly different?

35) Supplementary Fig. 2 Legend. Is the trace shown here a single measurement, or is it the mean of several ones? As a validation of the system, replicate traces or means with errors should be shown to illustrate the reproducibility of the system.

36) Supplementary Fig. 5 Legend. The goodness of fit test is not mentioned in the Supplementary Information.

37) Supplementary Fig. 7 Legend. Avoid the use of the term 'average' as this lacks specificity. Change to mean if this is what is meant.

38) Supplementary Fig. 7 Legend. The inserts in two of the graphs should be explained.

39) Supplementary Fig. 8 Legend. Was there an attempt to compare the radii across the strains? While the microcolony radii look different on the particle surface, does the inflection point differ between the two strains, and do the radii of the internal microcolonies differ?

40) Supplementary Fig. 9 Legend. Avoid the use of the term 'average' as this lacks specificity. Change to mean if this is what is meant.

Responses are in blue. Additions to the manuscript text are noted in **bolded** font. Line numbers refer to lines in the Word version with track changes on.

Reviewer #1 (Remarks to the Author):

The authors investigate the expression of Nir and Nar genes of *ps. aeruginosa* cells embedded in agar disks, immersed in microfluidics chambers. Fluoresce markers were inserted in the promotor region of genes, so that the gene expression could be visualized. To monitor oxygen the authors applied oxygen sensitive nanoparticles to their agar disks. With this setup, the authors performed time series experiments and could demonstrate a gradual development of expression levels of Nir and Nar inversely correlated with oxygen availability in the agar discs. The oxygen was consumed by *Ps. Aeruginosa* cells, themselves, and in this way, the cells triggered the evolution of sub oxic sites in the discs, thereby promoting the expression of Nir and Nar genes in cells reseeded in these environment. The NAR expression preceded the Nir expression in time and in the case of high nitrate concentrations also in intensity in full agreement with results from classical culture studies.

In general, I find this study very elegant designed and performed. The study shed light on the dynamics physical and biological mechanisms operating on a very small scale in particles. The work also offers an elegant technology, enabling direct in situ visualization of proxies for microbial metabolic processes. Data are convincing and the paper is well written. I recommend publication as is.

We thank the reviewer greatly for the thorough look at our work and the favorable comments regarding the design, execution, and impact of our work.

Reviewer #2 (Remarks to the Author):

Comments from the Reviewer of the Smriga et al. manuscript ‘Denitrifying bacteria respond to and shape microscale gradients within particulate matrices’ (Nat Comms)

Smriga et al. present a manuscript describing their investigations of the impact of oxygen levels on nitrate and nitrite reductase expression using the model pseudomonad, *P. aeruginosa* PA01 and artificial agarose particles. While the use of various polymers to capture bacteria into a 3-D structure and to test growth and phenotype response to changing conditions using microfluidics is not novel, I have found this a very interesting manuscript to read, and it adds to our greater understanding of how complex 3-D structures such as soil pore networks at the millimetre and micrometre scales effect bacterial growth and activity.

I would like three comments added to the appropriate places. First, a justification for using *P. aeruginosa* PA01 in this work needs to be added to the end of the Introduction; PA01 is not really a model for denitrifying bacteria in soils or sediments, though it is regarded as a model opportunistic human pathogen.

We appreciate the comments to better address the use of this model bacterium for its known metabolic abilities. We agree that *P. aeruginosa* PA01 is best known for its medical application as an opportunistic human pathogen, but it is indeed found in many natural environments as well. We note this more clearly in the last paragraph of the introduction as the reviewer recommends. Line 110:

“We employ genetically-engineered strains of the bacterium *Pseudomonas aeruginosa* wild-type PAO1. *P. aeruginosa* is found widely in the natural environment, across soils³¹, estuarine³², and marine systems³³. Further, it is an ideal model organism representing a ubiquitous genus of Proteobacteria. In addition to its environmental relevance, *P. aeruginosa* is most commonly known as an opportunistic human pathogen known to thrive in thick biofilms such as the cystic fibrosis lung, where it can denitrify³⁴.”

We also agree with the reviewer that the study has the goal to provide a broader understanding of spatial phenotypic variation in three-dimensional space under a defined and controlled diffusive landscape. Edited in Line 118: **“Our results reveal prominent gene expression spatial heterogeneity driven by microscale features and mechanisms that may be common for hydrated particle-based denitrification under varying bulk oxygen and nutrient regimes, which have cross-cutting utility for diverse systems.”**

Second, the flagella-mediated mobility, aero and chemotaxis responses of PAO1 need to be acknowledged and the impact this might have on nitrate and nitrate gene expression measurements within microcolonies considered (no independent measurements of cell densities or distributions were made, e.g. by using a third GFP-marked strain).

We appreciate the comment to acknowledge the potential for active motility in this system. In our experimental setting, the agarose concentration was chosen to avoid movement beyond the simple expansion of the colony driven by cell division. In other words, the matrix was tight enough as to preclude any aero- or chemotaxis. We now mention the lack of chemotactic potential throughout the manuscript, most notably in:

Line 148: **“Seeding bacterial cells within a 1% agarose matrix allows colony expansion due only to growth (passive movement) and prevents active aerotactic and chemotactic movement driven by flagellar motility³⁷.”**

and Line 376: **“Whereas aerotaxis and other chemotactic responses were not permitted due to the tightness of the agarose matrix, future studies could investigate how motile bacteria may disrupt the emergent gradients observed here.”**

We address additional aero/chemotaxis comments in further detail below.

Third, I would like to see some reflection on how the physical structure of the agarose particles might differ from particulate matter found in soils and sediments, and from the mucosa or EPS matrix of bacterial biofilms. Agarose particles will be uniformly hydrated and will not provide varied solid surfaces for attachment as compared to soil and sediment particles which may also contain their own pore network. The agarose particles are more similar to mucosa and biofilms, but at the agarose concentrations used, are likely to limit free cell movement and compress expanding microcolonies.

We thank the reviewer for noting the differences between porous solid and fully hydrated environments. We agree that our model system better relates to hydrated solid matrices rather than solid porous environments, and never intended to confuse the readers. We now clarify our meaning to be in reference to hydrated and three-dimensional mechanical confined space (e.g., EPS, tissue, or solid matrices) within these soil/sediment/particle systems, and further add an example regarding reef mucosa. Line 56: **“Many bacteria proliferate in surface attached microbial communities embedded in matrices of extracellular polymeric substances⁸. Moreover, soft aggregates in hydrated environments provide resources and microscale niches to sustain microbial communities and thus play a fundamental role to maintain key ecological processes. For instance, soils⁹, wastewater systems¹⁰, and marine ecosystems^{7,11} all contain ubiquitous surface**

attached microbial communities providing spatial niches that harbor denitrification in hydrated matrices¹².”

We further augment the text with Line 66: “In a marine context, recent evidence has shown that coral-generated mucus harbors active denitrifiers even among tropical reefs a few meters below the atmospheric interface ¹⁶. In several ecosystems, bacterial colonies grow subject to three dimensional mechanical confinement whereby a colony’s growth will modify its local physicochemical environment that may subsequently effect interactions with neighboring colonies¹⁷.”

I was minded that very few statistical tests results were mentioned in the text or figures to substantiate claims that there were differences in treatments. Some comments should be added to say when curves or other data were assessed qualitatively and when a statistical test was employed. Perhaps some summary tables of mean values, infection points and final values for microcolony growth and expression levels could be provided in the Supplementary Information along with simple statistical tests.

We appreciate the suggestion and apologize for the oversight, and now provide a more complete statistical description of our methods. We provided in every figure, captions and main text all the statistics required to comply with the journal guidelines, guide the readers, and address specifically the requests highlighted in the reviewer comments.

Additions were made to the following figures: In Figure 2 we provided the goodness of fit for each panel and subplot, in Supplementary Figure 1 we provided mean and standard deviation to be consistent throughout the text, in Supplementary Figure 2 we provided replicates, in Supplementary Figure 4 we provided goodness of fit for all the panels.

We further include the following additional text throughout the manuscript:

“ ... (one-way ANOVA, $F = 27.9$, $p = 4.0 \times 10^{-11}$ for Nar midpoints, $F = 14.9$, $p = 4.5 \times 10^{-8}$ for Nir midpoints, $F = 17.2$, $p = 9.5 \times 10^{-9}$ for Nar inflection points, and $F = 21.0$, $p = 1.0 \times 10^{-9}$ for Nir inflection points).” (Line 234).

“The midpoint of the slope (white diamonds) and the inflection point of the slope (circles) are indicated. NarK-GFP shows significantly higher expression between time points for midpoints (one way ANOVA; $p = 4.0 \times 10^{-11}$) and for inflection points (one way ANOVA; $p = 9.5 \times 10^{-9}$). NirS-dsRed also showed significantly higher expression between time points for midpoints (one way ANOVA; $p = 4.5 \times 10^{-8}$) and for inflection points (one way Anova; $p = 1.0 \times 10^{-9}$).” (caption, Figure 3.)

“In the aerobic zone, the median expression fraction of NarK-GFP is 3.1×10^{-5} (interquartile range 5.5×10^{-5}) whereas the NirS-dsRed expression fraction is 2.7×10^{-5} (interquartile range 1.5×10^{-5}). Expression of both genes are not significantly different from each other (Wilcoxon rank sum; $p = 0.1$). In the anaerobic zone, the median expression fraction of NarK-GFP and NirS-dsRed are 0.70 (interquartile 0.39) and 0.94, (interquartile 0.24), respectively. These anoxic fractional expressions are significantly different from each other (Wilcoxon rank sum; $p = 2.2 \times 10^{-32}$). The occurrence of heterogenous partially-on microcolonies reflects a sharp transition zone between presumptive aerobic and anoxic conditions ($x \sim 100 - 250 \mu\text{m}$). d, Probability density functions for each reporter strain indicate stronger bimodality and higher binary expression for NirS-dsRed than for NarK-GFP. The distribution of NarK v. NirS expression are significantly different from each other (two sample Kolmogorov-Smirnov test;

$p = 1.2 \times 10^{-40}$) with significantly different medians (Wilcoxon rank sum; $p = 1.9 \times 10^{-33}$).” (caption, Figure 5.)

“All treatments for both NarK-GFP and NirS-dsRed are significantly different from each other (One way ANOVA, $p < 0.001$).” (Caption, Supplementary Figure 1)

Comments

1) Lines 18-20. It is not clear what is meant by this sentence as modelling particles (and not bacterial distributions on or in them) is conflated with the experimental observations of bacterial growth within particles (and not of particle shapes).

We corrected the sentence and introduced the concept of gene expression spatial heterogeneity which was missing from the previous text.

2) Line 26. This should be altered to ‘nitrate and nitrite reductase expression’ or something similar.

We changed the sentence as suggested (Line 27).

3) Line 27. Correct to ‘develops’.

Corrected.

4) Introduction. When I read this, I had the overall impression that the manuscript was focussed on soils and sediments, rather than at human mucosa or biofilms, which makes *P. aeruginosa* PA10 a rather unusual choice for a test organism. Perhaps this needs to be justified towards the end of the Introduction (lines 81-83) rather than at the start of the results (lines 90-93).

We thank the reviewer for providing these suggestions to better inform the reader regarding the objectives of our research.

Line 61 changed: “solid” to “**hydrated matrices**”.

Line 115 we moved the following (edited) text from the start of the results to the end of the introductions. “**Similar studies³⁵ have used this organism to test the development of anoxia within model particles³⁶ and to report the biological response of nitrate and nitrite reductase expression³⁵. Like similar heterotrophic facultative denitrifiers, it respire nitrogen oxyanions (e.g., NO_3^- and NO_2^-) in lieu of O_2 under suboxic conditions.**”

Line 118 we added: “**Our results reveal prominent gene expression spatial heterogeneity driven by microscale features and mechanisms that may be common for hydrated particle-based denitrification under varying bulk oxygen and nutrient regimes, which have cross-cutting utility for diverse systems.**”

5) Details. The flagella-mediated motility of PA01 cells should be noted, as should be the chemotaxis and aerotaxis responses, as these will all play some role in the distribution of cells within microcolonies as local population densities increase. Will quorum sensing also be a factor?

To clarify the above we added the following text (Line 148): “**Seeding bacterial cells within a 1% agarose matrix allows colony expansion due only to growth (passive movement) and prevents**

active aerotactic and chemotactic movement driven by flagellar motility³⁷.” Our comment to Item #6 below additionally addresses a similar issue.

6) Line 98-100. Were any observations or tests to be made to determine whether the agarose-trapped cells were capable of moving within the agarose disks? PA01 cells are motile and can move in semi-solid agar (0.3% w/v), and cell motility will result in faster microcolony expansion; furthermore, a chemotaxis response will result in the hollowing-out of microcolonies when nutrients are reduced.

We thank the reviewer for bringing up such interesting questions regarding active movement. Per Burrows 2012, active movement in *P. aeruginosa*, such as flagellar swarming, occurs over a larger space and over a shorter period of time than our experiments ($\sim 1 \text{ mm h}^{-1}$). In addition, when bacterial cells display active movement, the resulting colony shapes change dramatically, diverging from the regular spheroidal shapes that we observe in our experiments. In our experiments we used an agarose concentration of 1% w/v, far higher than the classic “swimming agarose” usually employed to investigate active movement. Additionally, no motion was detected within individual microcolonies, indicating the cells were non-motile. In Line 148 we now provide a clarification on this specific topic: **“Seeding bacterial cells within a 1% agarose matrix allows colony expansion due only to growth (passive movement) and prevents active aerotactic and chemotactic movement driven by flagellar motility³⁷.”**

7) Line 111. Some reference to other work which defines anoxic conditions for PA01 should be provided, otherwise this statement needs to be reworded, perhaps using the expression of the nitrate and nitrite reductases as an indicator that anoxic conditions had been reached.

The text was moved to Line 115, now mentioning the nitrate and nitrite reductases explicitly: **“Similar studies³⁵ have used this organism to test the development of anoxia within model particles³⁶ and to report the biological response of nitrate and nitrite reductase expression³⁵.”**

8) Line 118. Anoxia is not ‘initiated’ but rather first reached or achieved, and the anoxic region expanded, rather than anoxia itself.

We agree with the reviewer and we changed each time we used the term ‘initiated’ with ‘achieved’.

Line 158 changed the term ‘initiated’ with ‘**manifested.**’

Line 162 changed the term ‘initiated’ with ‘**rose first.**’

Line 846 changed the term ‘initiated’ with ‘**occurred.**’

9) Line 142. Avoid the use of the term ‘averaged’ as this lacks specificity. Change to mean if this is what is meant.

We thank the reviewer for suggesting we be more precise. We changed the term ‘averaged’ to ‘**mean**’ at each location in which was used.

10) Line 142. What are the errors indicated here?

The values indicated are standard deviations (**sd**). We had not annotated all of the “ \pm ” for simplicity in style, but we now do for clarity.

11) Line 142. This is the first place where means and errors are provided, but why not earlier in the descriptions of microcolony sizes and anoxic zones?

Each mean we report has an associated standard deviation. In the initial paragraphs of the results we opt to more qualitatively describe the results in order to provide a narrative description of our results. The text points the reader to figures where the data are displayed.

12) Lines 147-148. Please reconsider this sentence. Agarose particles were continuously washed with LB media, and cells within particles are surrounded by fluid too (but not always oxygenated) – so which is the bulk fluid?

In the context of this sentence, we refer to millifluidic experiments that maintained continuous flow of either 50% air or 100% nitrogen-equilibrated LB media. Here the “bulk” fluid refers to the fluid external to the agarose particle.

13) Lines 152-153. Is it relevant to mention here that some of the particles were sacrificed for measurements? This should be in the Methods.

We agree with the reviewer that this is extraneous information and changed the following text: “Following an initial incubation period, particles were sacrificed over time, which included imaging of the O₂ profile via co-embedded nanosensors.” to “**A subset of particles were co-embedded with oxygen nanosensors to enable imaging of the O₂ landscape.**” (Line 197). We felt it necessary to point out that only some particles contained these O₂ sensing particles.

14) Line 181. I am not sure that the equation is required here and is probably more suited to the Supplementary Information.

We appreciate the reviewer’s perspective, but we respectfully would prefer to keep the equation in the main text, as the functional form helps guide the interpretation of regulatory response to both oxygen inhibition and nitrate stimulation.

15) Lines 185 – 187. According to the Journal Guidelines, the Degrees of Freedom should be provided for each F statistic.

We thank the reviewer for highlighting the Journal Guidelines requirements. We have gone through the paper and figures to provide the necessary tests as indicated by the reviewer and to comply with these guidelines. We provide additional information regarding the statistical tests in detail below.

16) Lines 195-197. The lower levels of Nir expression within particles might be conflated by aero or chemotaxis of cells.

The reviewer is correct that in a system that permits aerotaxis and chemotaxis, this is an essential consideration, but these are not applicable to our setup. Please see responses above.

17) Line 208. Change to ‘as shown (Fig. 5b-c)’ or something similar.

We changed as suggested.

18) Line 212. What is ‘IQR’? This not explained here or in the Supplementary Information.

We replaced as “interquartile range.”

19) Line 215. What is the K-S test? This should be indicated in full or the acronym noted in the Supplementary Information.

Kolmogorov-Smirnov which refers to a non-parametric test to compare the similarity of one-dimensional probability distributions.

20) Line 285. *P. aeruginosa* PA01 is more than an animal pathogen.

Indeed the reviewer is correct, and we point out that it is found broadly across environments: **“We employ genetically-engineered strains of the bacterium *Pseudomonas aeruginosa* wild-type PA01. *P. aeruginosa* is found widely in the natural environment, across soils³¹, estuarine³², and marine systems³³. Further, it is an ideal model organism representing a ubiquitous genus of Proteobacteria. In addition to its environmental relevance, *P. aeruginosa* is most commonly known as an opportunistic human pathogen known to thrive in thick biofilms such as the cystic fibrosis lung, where it can denitrify³⁴. Similar studies³⁵ have used this organism to test the development of anoxia within model particles³⁶ and to report the biological response of nitrate and nitrite reductase expression³⁵.”** (Lines 110)

21) Lines 307-309. Surely 3-D diffusion centred on microcolonies within the disks occurred?

The reviewer makes an important point. We are referring to the larger-scale diffusion, but there are microgradients that surely manifest centered on the colonies themselves. We have changed the sentence to read: **“Here, the agarose discs permitted lateral diffusion from the bulk and smaller-scale three-dimensional diffusion across a colony, as may be common for ventilation in natural particle systems, such as biofilms, microbial mats, sediments, and marine aggregates¹³.”** (Line 362)

22) Lines 307-309. The comment regarding 1-D diffusion in biofilms is probably not defensible. While there is no agreed minimal depth (in terms of distance or the number of cell layers), even a mono-layer will establish gradients immediately above it in the laminar layer between the cells and the bulk fluid.

We agree with the reviewer and edited the sentence as in Comment #21 above.

23) Line 310. It is unlikely that a steady boundary condition was maintained around the disks, as the downstream region would be subject to turbulence resulting from differential liquid pressure/flow.

The fluid velocities applied in these experiments created a laminar flow regime, thus permitting stable boundary conditions around the particle with negligible turbulence. During validation tests of the system with tracer particles, we visually confirmed the occurrence of streamlines with negligible turbulence in the shadow of the particle

24) Lines 316-317. Cell densities within microcolonies were not measured in this work, so care must be taken here and elsewhere to acknowledge that nitrate and nitrate reductase expression may not be directly linked to cell densities; aero and chemotaxis may have resulted in cells migrating out of the colony cores towards the edges.

We very much appreciate this insight. As described above, we worked under regimes that would not allow active movement. The reviewer does stimulate an exciting question, however, that other systems and future experiments could consider how aero- and chemotaxis may alter these results. We do comment at the end of the manuscript in this regard: **“Whereas aerotaxis and other chemotactic responses were not permitted due to the tightness of the agarose matrix, future studies could investigate how motile bacteria may disrupt the emergent gradients observed here.”** (Line 376).

25) Supplementary Methods. Where all assays of cell growth and behaviour undertaken at 37°C?

Everything was conducted at room temperature, to clarify we added this sentence: **“All millifluidic experiments were conducted at room temperature (21 ± 1 °C).”** (Line 811).

26) Lines 641-645. A comment should be added in this section to indicate that the promoter fusions had no significant effect on the down-stream expression of the nitrate and nitrite reductase genes.

We thank the reviewer for indicating this specific information and we correct the text adding the following sentence: **“The promoter fusion used in this study does not influence the down-stream expression of the reductase genes.”** (Line 754).

27) Line 648. Regrettably both the naming and composition of LB has become a little confused, and the level of NaCl in different formulations is known to effect gene expression in Escherichia coli and other bacteria. I would suggest adding the formulation of LM here in parentheses or citing the supplier if pre-mixed media was used.

We agree with the reviewer, we provided the specific information of the provider: **“BD Difco™ LB Broth (Miller), product number 244620”** (Line 758)

28) Lines 671-673. What temperature was the molten agar allowed to cool to before cells were added? Normally the molten agar would be equilibrated to 50/55°C (just above setting temperature) before use.

We always measured the temperature of the low melting agarose to ensure it was at ~40°C. Low melting (rather than standard) agarose is more conducive to seeding with live bacteria. We explicitly mention temperatures in Line 781: **“When the molten agarose had cooled (~40 °C), cells were mixed into it at various volumes, e.g., for a cell diluent with $OD_{600} = 0.16$, the addition of 133 μ L of cells to 10 mL of molten agarose achieved a seeding density of $\sim 10^6$ cells mL^{-1} .”**

29) Lines 679 etc. It is annoying to have to report Imperial measurements, but here and elsewhere when it is necessary, the appropriate SI conversions should be provided.

We agree that all data should (and do) follow SI conventions, but we maintain the Imperial units for the specific materials sold in imperial units from McMaster-Carr that we used for ease of reproduction of our results and system by others. We opt not to provide a 1 in = 25.4 mm conversion.

30) No section is provided in the Supplementary Information detailing the approach taken to data analysis and comparisons. What software was used to fit and test curves? What statistical software was used for tests? Were data or residuals (e.g., from the ANOVA tests) assumed to be Normally distributed or was this tested? (Journal Guidelines require that assumptions are noted).

We added the following text in the methods section: **“Statistical tests. All statistical tests were conducted in Matlab (R2019a) using parametric tests in the case of normally distributed data; otherwise non-parametric tests were used. The ANOVA tests were conducted for normally distributed data considering p -values < 0.05 to be statistically significant, although the resulting p -values were generally < 0.001 . We report the goodness of fit as well as sample sizes throughout.** (Line 926).

31) Fig. 1 Legend. The scale bar in (b) is indicates as approx. 1500 μ m and not as 1.2 cm as stated.

We have corrected this.

32) Fig. 5. The graphs might be more easily interpreted if the outliers were removed. Are the outliers incorporated into the density functions?

We deleted the outliers from the figure, but the outliers remain included when calculating the probability density functions.

33) Fig. 5. The medians are very hard to see and perhaps could be indicated by thicker lines.

We made the medians more visible by changing them to blue crosses.

34) Supplementary Fig. 1 Legend. If mean signals are discussed, why a box-plots shown (normally indicating min, Q2, median, Q3 and max data)? Why are no statistical tests shown indicating whether or not treatments are significantly different?

We now show requested statistics and changed the box-plot to mean value and error bar and provided the appropriate statistical test for testing differences between treatment as suggested by the reviewer.

35) Supplementary Fig. 2 Legend. Is the trace shown here a single measurement, or is it the mean of several ones? As a validation of the system, replicate traces or means with errors should be shown to illustrate the reproducibility of the system.

This supplementary figure is now including replicates traces to show the dynamic of mean nanoparticle fluorescence and its variance among replicates thus confirming consistency of oxygen dynamic across replicates.

36) Supplementary Fig. 5 Legend. The goodness of fit test is not mentioned in the Supplementary Information.

We now provide for all the figures and panels the goodness of fit as requested.

37) Supplementary Fig. 7 Legend. Avoid the use of the term ‘average’ as this lacks specificity. Change to mean if this is what is meant.

We changed ‘average’ to ‘**mean**’.

38) Supplementary Fig. 7 Legend. The inserts in two of the graphs should be explained.

We now explicitly explain the insets are intended only to zoom into the region of variability for those panels while maintaining a consistent scale across the panels.

39) Supplementary Fig. 8 Legend. Was there an attempt to compare the radii across the strains? While the microcolony radii look different on the particle surface, does the inflection point differ between the two strains, and do the radii of the internal microcolonies differ?

We appreciate the reviewer’s observations here. The microcolony radii may differ across particle regions owing to slight differences in seeding density and/or particle size (indicated by ‘distance to particle edge’). This figure supports the following sentence in the main text: **“In this manner, the fluorescence signal diminishes after achieving its peak and microcolonies remain small behind the advancing fluorescence wave while colonies continue to expand ahead of it (Supplementary Fig. 5, Supplementary Figure 8).”** This sentence will be true even if the reporter strains statistically differed in colony radii in these experiments.

40) Supplementary Fig. 9 Legend. Avoid the use of the term ‘average’ as this lacks specificity. Change to mean if this is what is meant.

We changed ‘average’ to ‘**mean**’.

REVIEWERS' COMMENTS:

Reviewer #2 (Remarks to the Author):

Comments from the Reviewer of the Smriga et al. manuscript 'Denitrifying bacteria respond to and shape microscale gradients within particulate matrices' (Nat Comms)

Smriga et al. present their revised manuscript describing the impact of oxygen levels on nitrate and nitrite reductase expression in *P. aeruginosa* PA01 using artificial agarose particles. The Authors have responded well to the Reviewer's comments, including those of my own.

I feel that the reasoning behind using PA01 as a model organism in this work is far better presented and the other changes made at the start of the manuscript provide a better introduction to the investigations carried out in this work. It was important to note that PA01 cells are motile but that the agarose concentration used here inhibits this. I am happy with the statistical analysis of the experimental data, but note again that according to the journal's rules, that the test statistic plus degrees of freedom should be provided every time a statistical test is referred to in the Results and figure legends.

- 1) Line 107. Perhaps a comment could be made here in parentheses to note that habitats become even more heterogeneous if pore networks (between and within particles) is partially-saturated.
- 2) Line 197. Change 'were' to 'was'.
- 3) Lines 234-236. According to the Journal Guidelines, the Degrees of Freedom should be provided for each F statistic (e.g. $F_{1,3} = 27, P = 0.03$).
- 4) Lines 267-268. The Kolmogorov-Smirnov test statistic (D_n) plus the degrees of freedom should be provided.
- 5) Lines 577-580. ANOVA F statistics and degrees of freedom should be provided.
- 6) Lines 605-608. Wilcoxon (Rank sums) test statistics (W) and degrees of freedom should be provided.
- 7) Lines 612-613. The Kolmogorov-Smirnov test statistic (D_n) and Wilcoxon (Rank sums) test statistic (W) and degrees of freedom should be provided.
- 8) Line 637. ANOVA F statistics and degrees of freedom should be provided.
- 9) Lines 926-929. The statistics section should indicate what tests were used when the data (or residuals) were found not to be Normally distributed, and what tests were used to determine goodness of fit.

Response to Reviewers, second round. Responses are in blue.

Reviewer #2 (Remarks to the Author)

Comments from the Reviewer of the Smriga et al. manuscript ‘Denitrifying bacteria respond to and shape microscale gradients within particulate matrices’ (Nat Comms)

Smriga et al. present their revised manuscript describing the impact of oxygen levels on nitrate and nitrite reductase expression in *P. aeruginosa* PA01 using artificial agarose particles. The Authors have responded well to the Reviewer’s comments, including those of my own.

I feel that the reasoning behind using PA01 as a model organism in this work is far better presented and the other changes made at the start of the manuscript provide a better introduction to the investigations carried out in this work. It was important to note that PA01 cells are motile but that the agarose concentration used here inhibits this. I am happy with the statistical analysis of the experimental data, but note again that according to the journal’s rules, that the test statistic plus degrees of freedom should be provided every time a statistical test is referred to in the Results and figure legends.

We are pleased by the reviewer’s response in terms of our addressing reviewer comments and justifying PA01. We apologize we do not see where degrees of freedom are required in the journal rules, but supply these as appropriate in the results and figure legends as requested. We note all sample sizes are well in excess of 1,000.

1) Line 107. Perhaps a comment could be made here in parentheses to note that habitats become even more heterogeneous if pore networks (between and within particles) is partially-saturated.

We have added the sentence: “**The resulting heterogeneities can be exacerbated when pore networks are partially saturated with water.**” (Line 76)

2) Line 197. Change ‘were’ to ‘was’.

Corrected, thank you.

3) Lines 234-236. According to the Journal Guidelines, the Degrees of Freedom should be provided for each F statistic (e.g. $F_{1,3} = 27$, $P = 0.03$).

Done.

4) Lines 267-268. The Kolmogorov-Smirnov test statistic (D_n) plus the degrees of freedom should be provided.

We added D_n for these KS tests, but degrees of freedom is not necessarily appropriate with this test. Instead we have reported the sample size each of the two distributions contains.

5) Lines 577-580. ANOVA F statistics and degrees of freedom should be provided.

Done.

6) Lines 605-608. Wilcoxon (Rank sums) test statistics (W) and degrees of freedom should be provided.

We added W for these tests, but degrees of freedom is not necessarily appropriate with this test. Instead we have reported the sample size each of the two distributions contains.

7) Lines 612-613. The Kolmogorov-Smirnov test statistic (Dn) and Wilcoxon (Rank sums) test statistic (W) and degrees of freedom should be provided.

We added Dn/W for these tests, but degrees of freedom is not necessarily appropriate. Instead we have reported the sample size each of the two distributions contains.

8) Line 637. ANOVA F statistics and degrees of freedom should be provided.

Done.

9) Lines 926-929. The statistics section should indicate what tests were used when the data (or residuals) were found not to be Normally distributed, and what tests were used to determine goodness of fit.

The *Statistics and Reproducibility* section of the methods has been expanded to state: “All statistical tests were conducted in Matlab (R2019a) using parametric tests in the case of normally distributed data; otherwise non-parametric tests were used. The ANOVA tests were conducted for normally distributed data considering p -values < 0.05 to be statistically significant, although the resulting p -values were generally < 0.001 . Denitrification gene intensity was fit using the Matlab curve fitting toolbox (Supplemental) and colony sizes using Matlab nonlinear regression assuming an exponential of the form, $Size = k_1 e^{-k_2 r}$.”